# Parent Fruit and Vegetable Consumption Outcomes from the Translational ‘Time for Healthy Habits’ Trial: Secondary Outcomes from a Partially Randomized Preference Trial

**DOI:** 10.3390/ijerph19106165

**Published:** 2022-05-19

**Authors:** Rebecca J. Wyse, Jacklyn K. Jackson, Megan L. Hammersley, Fiona Stacey, Rachel A. Jones, Anthony Okely, Amanda Green, Sze Lin Yoong, Christophe Lecathelinais, Christine Innes-Hughes, Joe Xu, Karen Gillham, Chris Rissel

**Affiliations:** 1School of Medicine and Public Health, College of Health, Medicine and Wellbeing, University of Newcastle, University Drive, Newcastle, NSW 2308, Australia; jacklyn.jackson@health.nsw.gov.au (J.K.J.); fiona.stacey@health.nsw.gov.au (F.S.); serene.yoong@health.nsw.gov.au (S.L.Y.); christophe.lecathelinais@health.nsw.gov.au (C.L.); 2Hunter New England Population Health, Wallsend, Newcastle, NSW 2287, Australia; karen.gillham@health.nsw.gov.au; 3Hunter Medical Research Institute, New Lambton Heights, Newcastle, NSW 2305, Australia; 4Priority Research Centre for Health Behaviour, University of Newcastle, University Drive, Newcastle, NSW 2308, Australia; 5Early Start, Faculty of the Arts, Social Sciences and Humanities, University of Wollongong, Wollongong, NSW 2522, Australia; mhammers@uow.edu.au (M.L.H.); rachelj@uow.edu.au (R.A.J.); tokely@uow.edu.au (A.O.); 6School of Health and Society, Faculty of the Arts, Social Sciences and Humanities, University of Wollongong, Wollongong, NSW 2522, Australia; 7Illawarra Health and Medicine Research Institute, Wollongong, NSW 2522, Australia; 8School of Education, Faculty of the Arts, Social Sciences and Humanities, University of Wollongong, Wollongong, NSW 2522, Australia; 9Centre for Population Health, New South Wales Ministry of Health, St. Leonards, Sydney, NSW 2065, Australia; amanda.green@health.nsw.gov.au (A.G.); christine.inneshughes@health.nsw.gov.au (C.I.-H.); jx1158@gmail.com (J.X.); 10School of Health Sciences, Department of Nursing and Allied Health, Swinburne University of Technology, Hawthorn, Melbourne, VIC 3122, Australia; 11College of Medicine and Public Health, Flinders University, Darwin, NT 0800, Australia; chris.rissel@flinders.edu.au

**Keywords:** research translation trial, RCT, parents, partially randomized preference trial, obesity prevention, fruit and vegetable intake, telephone-based intervention, online intervention

## Abstract

Healthy eating and active living interventions targeting parents of young children could have benefits for both children and parents. The aim of this study was to assess the effectiveness of two remotely delivered healthy eating and active living interventions delivered at scale to parents, in increasing parent vegetable and fruit consumption (pre-specified secondary outcomes). Parents of children aged 2–6 years residing in New South Wales, Australia (n = 458), were recruited to a partially randomized preference trial consisting of three groups (telephone intervention (n = 95); online intervention (n = 218); written material (Control) (n = 145)). This design allowed parents with a strong preference to select their preferred intervention, and once preference trends had been established, all parents that were subsequently recruited were randomized to obtain robust relative effects. Parent vegetable and fruit consumption was assessed via telephone interview at baseline and 9 months later. At follow-up, randomized parents who received the telephone intervention (n = 73) had significantly higher vegetable consumption compared to those who received the written control (n = 81) (+0.41 serves/day, *p* = 0.04), but there were no differences in parents allocated to intervention groups based on preference. No differences in fruit consumption were found for randomized or preference participants for either the telephone or online intervention. There may be some benefit to parents participating in the Healthy Habits Plus (telephone-based) intervention aimed at improving the eating behaviors of their children.

## 1. Introduction

Fruits and vegetables are essential components of a healthy diet [1], and are important determinants of health, development and wellbeing across the lifespan [2]. Internationally, it is recommended adults consume a minimum of five fruit and vegetable servings per day (approximately 400 g) to promote good health [3]. However, at a population level, the minimum targets for fruit and vegetable intakes are rarely met across all age groups [2]. As low fruit and vegetable consumption is an important risk factor for several chronic diseases, including some cancers [3,4], coronary heart disease [5] and stroke [6], exploration of strategies that may improve population fruit and vegetable intakes remains a public health priority [7,8].

Early childhood is an important period to prioritize good nutrition [9,10] as the eating habits adopted by children early in life track throughout the lifespan and into adulthood [11,12,13]. Parents and carers (hereafter ‘parents’) of young children play a critical role in creating a supportive home food environment through the foods that they make available and accessible to their children [14] and their role modeling of healthy eating behaviors [15]. Parent-targeted interventions which focus on changing the home food environment of young children are effective in establishing healthy child dietary habits [15,16], and could be an important public health strategy to minimize the future burden of diet-related chronic diseases.

Although face-to-face parent nutrition education programs are an effective, feasible and acceptable method of intervention to improve child fruit and vegetable intakes [16], barriers including travel [17], cost [18], childcare requirements [19] and limited time to attend such programs [18] limit their potential to impact public health nutrition. Online and telephone-based interventions can overcome such barriers, and offer greater convenience and flexibility for parents, supporting wide-scale participation [16]. Interventions delivered via these modalities are supported by systematic review evidence, suggesting that tailored telephone- or computer-based interventions represent effective alternatives to face-to-face interventions for increasing fruit and vegetable intakes [20,21].

Child lifestyle interventions aimed at engaging only parents can be more effective than those aimed at both the parent and child together [22], as parents have been shown to be the most important ‘agents of change’ for influencing child lifestyle behaviors. This has been shown to be especially important for fruit and vegetable consumption, with research consistently demonstrating that parental modeling, preferences and intake are positively associated with children’s food preference, as well as children’s intake of fruit and vegetables [23,24,25].

In addition, parents’ own behaviors may be modified by the shared environment they establish for their children, as interventions that focus on changing the home food environment may impact the dietary behaviors of parents as well as their children. For example, ‘High 5 for Kids’ tested the effectiveness of a (face-to-face) home-based intervention including four 60 min home visits to teach parents how to provide a positive fruit and vegetable environment for their preschool child [26]. Evaluation of the ‘High 5 for Kids’ found a significant increase in intervention parent fruit and vegetable intakes (+0.20 serves, *p* = 0.05), compared with the usual care control [26]. Changes in parents’ intake of fruit and vegetable intakes was also found to be a significant predictor of their children’s intake of fruit and vegetables. This finding is consistent with the ‘Healthy Habits’ randomized controlled trial, where parents of children aged 2–5 years received four telephone calls to improve key characteristics of the home food environment. Parents and children that received the ‘Healthy Habits’ intervention were found to have significantly higher fruit and vegetable intakes at the 12-months and 18-months follow-up compared with control participants [27,28]. Such findings highlight the potential co-benefits of parent-directed child healthy lifestyle interventions. 

‘Time for Healthy Habits’ was a translational trial that recruited parents of children aged from 2 to 6 years, offered as a free population-wide service across New South Wales (NSW), Australia, to mimic real-world implementation. ‘Time for Healthy Habits’ builds upon two remotely delivered (telephone or online) and efficacious interventions [28,29] to explore their effectiveness when offered to the community through integration into existing preventive health services. Currently, there is limited real-world evidence to gauge the likely impact of such a service on public health nutrition. The ‘Time for Healthy Habits’ trial sought to address this gap through a parallel partially randomized preference trial comprising three arms: (1) a telephone-based intervention (Healthy Habits Plus); (2) an online intervention (Time2bHealthy) and (3) a control arm, which consisted of printed educational materials. The current study aimed to evaluate the effectiveness of the telephone and online interventions in changing parent fruit and vegetable consumption at 9 months post-baseline, a secondary outcome of the ‘Time for Healthy Habits’ trial.

## 2. Materials and Methods

### 2.1. Study Sample 

Time for Healthy Habits’ employed a partially randomized preference study design, and detailed description of these methods have been published elsewhere [30]. Briefly, parents of children aged from 2 to 6 years, residing in New South Wales, Australia, were recruited. The trial design allowed parents who had a strong preference to elect their preferred group (telephone, online or written (control)) (i.e., participants were asked: ‘Do you have a strong preference for the way in which you receive healthy lifestyle advice or support about your child’? If they responded ‘yes’, they were asked ‘Would you prefer to receive healthy lifestyle advice or support via written information, telephone or online’). Parents who did not have a strong preference were randomly allocated. Additionally, a stopping rule was applied to the preference arm to ensure enough participants were randomized to different groups to power the primary analysis [30]. Participants in the randomized arm were randomly allocated using a random number function generated by an independent statistician, whereby each of the three arms were given a 33% likelihood of being selected. Partially randomized preference trials offer greater external validity than traditional randomized controlled trials (RCTs), and have been recommended to test the effectiveness of interventions in real-world settings, especially if the control receives usual care [31]. Following baseline data collection (and prior to the application of the stopping rule), participants were given the option to choose the intervention which they received (preference arm), or allocated via randomization to one of the three groups (randomized arm). After the application of the stopping rule, all participants were randomized.

The trial (including the secondary outcomes reported in the current study) was registered with the Australian New Zealand Clinical Trials Registry on 12 March 2019 (ACTRN12619000396123). Ethics was approved by the South Western Sydney Local Health District Human Research Ethics Committee (HE18/300) in conjunction with site specific approvals by Murrumbidgee Local Health District (LHD) Human Research Ethics Committee (HREC), Hunter New England LHD HREC, Illawarra Shoalhaven LHD HREC, Southern New South Wales LHD HREC and South Eastern Sydney LHD HREC, as well the University of Newcastle HREC (H-2019-0188) and the University of Wollongong HREC (HE2019/207). The pre-registered primary and secondary outcomes for this trial relating to child dietary intake and movement behaviors have been reported elsewhere [32]. The current study reports on pre-registered outcomes related to parent intake of fruit and vegetables. 

### 2.2. Participant Recruitment

Recruitment was led by five participating LHDs and was conducted between April 2019 and April 2020. Recruitment officers from each LHD conducted face-to-face visits to sites that provided services to young children, including: Early Childhood Education and Care services, playgroups, clinics, library groups and other early childhood activities. The study was also promoted by the LHD health professionals who were asked to refer appropriate parents to the study. Information materials and flyers were distributed through local newsletters, community noticeboards, media releases, social media and local networks. Parents provided informed consent through hard-copy paper forms or online via the study website. 

### 2.3. Interventions

This trial tested two different intervention packages, one primarily delivered via the telephone, and one delivered online. While the interventions had the same aim (i.e., to improve child fruit and vegetable consumption) and there were similarities between the strategies employed and the intervention content; these were separate intervention packages. The two interventions, ‘Healthy Habits Plus’ and ‘Time2bHealthy’, have been previously described in detail elsewhere [30], and are summarized in Table 1. Both interventions were delivered remotely (via telephone or online) and provided practical information and tips to parents to support child healthy eating, and movement behaviors (i.e., physical activity, screen time and sleep). Both interventions were intended to be completed over a 12-week period, and intervention components were underpinned by theory and behavior change techniques, including barrier identification, goal setting and self-monitoring. 

Healthy Habits Plus (hereafter, also referred to as the ‘telephone’ intervention) is a telephone-based program, consisting of six 20–30 min scripted support calls undertaken fortnightly to parents. Calls were delivered by trained para-professionals using computer-assistant telephone interview (CATI) software. Printed, hardcopy resources (i.e., a guidebook and menu planner) were mailed to participants to be used during and in between calls to facilitate behavior change. The telephone-based intervention draws on socio-ecologic theory [33] and the conceptual family-based intervention model of Golan and Weizman [34], which focuses on making changes to the home environment relating to access to healthy foods and opportunities to be active, creating supportive family routines and encouraging positive parental role-modeling. The telephone-based intervention content builds on the earlier version (Healthy Habits [28]) which focused solely on increasing fruit and vegetable consumption, to now include movement behaviors (physical activity, screen time and sleep). 

Time2bHealthy (hereafter, also referred to as the ‘online’ intervention) is an online-based program consisting of six modules, each taking approximately 30 min to complete. Participants were encouraged to complete one module each fortnight and received weekly email reminders to log into the program. The online-based intervention also incorporated an optional closed Facebook group (moderated by a health professional), which allowed parents to communicate and share ideas regarding the implementation of healthy lifestyle changes in their family. The online-based intervention was underpinned by Social Cognitive Theory [35], which proposes that there are three influences on behavior: personal, behavioral and environmental, and that individuals learn from observing others. This theory also encompasses self-efficacy and proposes that individuals that have belief in their capabilities are more likely to achieve their goals. Intervention components such as goal setting and action planning, barrier identification, videos, activity planners and communication with the Facebook group were used to align the content to theory.

### 2.4. Control Group

Control participants received written educational materials developed by the NSW Office of Preventive Health, outlining healthy eating and movement behavior recommendations for young children. Parents were emailed or mailed (depending on their preference) two factsheets every fortnight and received a 12-page summary booklet at the end. Fact sheet topics included: healthy breakfasts; healthy lunchboxes; positive family mealtime; healthy drinks; fussy eating; encouraging children to eat vegetables and fruit; healthy snacks; active play; reducing screen time; and encouraging healthy sleep habits.

Parent role modeling of healthy eating behaviors (i.e., parents sit down as a family, and eat the same food they feed to their child and show the child how much they enjoy healthy food), was a common practical tip outlined within the written fact sheets. Parents were also prompted to access relevant NSW government websites, including healthykids.nsw.gov.au (accessed on 13 April 2022) (i.e., Munch & Move, an NSW Health initiative that supports the healthy development of children (0–5 years)), and raisingchildren.net.au (i.e., an Australian parenting website). 

### 2.5. Data Collection and Measures

All data collection occurred via telephone by trained interviewers. The survey contained standardized scripted items and was administered via CATI. Participating parents were called at baseline (prior to group allocation between May 2019 and May 2020) and approximately 6 months post-intervention (i.e., 9 months post-baseline, between March 2020 and March 2021). 

Items assessing demographic characteristics of participating parents (i.e., age and sex) were included. Additional demographic items to assess parent highest level of education/qualification completed, annual household income, and Aboriginal or Torres Strait Islander identification were based on questions from the NSW Population Health Survey [36].

Items to assess the average daily serves of fruits and vegetables consumed by parents (main outcome) were sourced from the Australian National Nutrition Survey [37]. To assess parent vegetable intake, participants were asked: ‘How many serves of vegetables do you usually eat each day? One adult serve is a ½ cup of cooked vegetables or 1 cup of salad vegetables’. To assess parent fruit intake, participants were asked: ‘How many serves of fruit do you usually eat each day? An adult serve is 1 medium piece or 2 small pieces of fruit or 1 cup of diced pieces’. Responses to these survey items have been previously found to be significantly associated with biomarkers for fruit and vegetable intake (alpha- and beta-carotene and red-cell folate) in a sample of 1598 Australian adults [38]. Further, in a sample of 100 Australian adults, these short fruit and vegetable screener questions were found to have good agreement with intakes reported using 24 h recalls [39]. 

### 2.6. Statistical Analysis

Parent intakes of fruit and vegetables were analyzed as separate outcomes using linear regression models. The appropriateness of linear regression models for this analysis was determined by examining the residuals’ normality through histograms and Q plots. Separate linear regressions were conducted to compare the effect of the telephone intervention vs. control, and online intervention vs. control, on parents’ mean daily fruit and vegetables serves. Consistent with the main outcomes analysis previously published [32], (i) the main analysis for the current study was conducted on randomized participants. Separate analyses were also completed for (ii) preference participants, and (iii) all participants in the data set. For each regression, an interaction term (‘group’ (interventions vs. control) by ‘arm’ (randomized vs. preference)) was added to the model. Using an intention-to-treat approach, we conducted (a) complete case analysis (i.e., including data for participants that had complete baseline and follow-up data), and (b) multiple imputations analysis (i.e., participants missing follow-up data had follow-up values imputed). All analyses were adjusted for baseline fruit and vegetable intakes. Analyses were conducted using SAS software, version 9.3, SAS Institute Inc., Cary, NC, USA.

## 3. Results

Of the 458 parents who completed baseline data collection, most were female (96.3%), university-educated (70.3%), and had an average annual household income over AUD 100,000 per year (68.4%) [32]. The average age of parents at baseline was 36.13 years (SD = 4.92) (See Table 2).

A total of 244 parents were given the option to choose their modality of intervention delivery, and 218 parents (89%) expressed a strong preference, with 22 parents (10%) choosing the telephone intervention, 132 parents (61%) choosing the online intervention and 64 parents (29%) choosing the written material (i.e., active control). Prior to the stopping rule being introduced, 26 parents did not express a strong preference and were randomly allocated to an intervention. After the application of the stopping rule, 214 participants were recruited and randomly allocated to an intervention. In the randomized arm (i.e., those randomized before and after the application of the stopping rule, n = 240), 73 parents (30%) were randomized to the telephone intervention, 86 (36%) were randomized to the online intervention, and 81 parents (34%) were randomized to the control [32]. 

Approximately 70% of randomly allocated parents (telephone: 62%; online: 62%; control: 76.5%) and 64% of parents allocated based on preference (telephone: 82%; online: 58%; control: 69%) provided 9-month post-baseline follow-up data (Figure 1). The participant characteristics by preference arm and group are shown in Table 2.

### 3.1. Vegetable Consumption

Randomized participants: The main analysis of the 240 randomized parents found that, at follow-up, parents randomized to the telephone group consumed significantly more vegetable serves than parents randomized to the control group (complete case: effect size 0.48 serves/day (95% CI: 0.07 to 0.88), *p* = 0.02; imputation: effect size 0.41 serves/day (95% CI: 0.02 to 0.81), *p* = 0.04) (See Table 3). There was no statistically significant difference between parents randomized to the online group and the control group (Table 3).

Preference participants: Analysis of the 218 parents allocated to their group of preference found no statistically significant differences between the control group and either the telephone group or online group (Table 3).

All participants: Analysis of all 458 parents found that there was no difference in the vegetable consumption of parents receiving the telephone or online intervention compared with control (Table 3).

### 3.2. Fruit Consumption

Randomized participants: The main analysis of randomized parents found no statistically significant difference at follow-up between the control group and either telephone group (complete case: effect size 0.08 serves/day (95% CI: −0.25 to 0.42), *p* = 0.62; imputation: effect size −0.05 serves/day (95% CI: −0.38 to 0.27), *p* = 0.75) or online group (complete case: effect size −0.16 serves/day (95% CI: −0.46 to 0.15), *p* = 0.32; imputation: effect size −0.02 serves/day (95% CI: −0.37 to 0.34), *p* = 0.92) (Table 4). 

Preference participants: The analysis of the parents allocated to their group based on preference found there were no statistically significant differences between control parents’ fruit consumption compared with either the telephone group or online group (Table 4).

All participants: Analysis of all parents indicated that there was no difference in the fruit consumption of parents receiving the telephone intervention or the online intervention compared with the control group (Table 4). 

## 4. Discussion

This study describes the changes in parent fruit and vegetable consumption at follow-up, a secondary outcome of the ‘Time for Healthy Habits’ trial. This translational trial found a statistically significant difference in the consumption of vegetables among parents randomly allocated to the telephone intervention (equivalent to increases of 30–36 g/day, based on a 75 g vegetable serve), but significant differences were not found for preference participants, or for fruit consumption. These findings suggest that children’s healthy lifestyle interventions delivered to parents via telephone can have broader public health nutrition impacts, highlighting the potential co-benefits for parent and child dietary intakes [32].

The changes in parent vegetable consumption in those randomized to the telephone group are consistent with the findings of the original ‘Healthy Habits’ intervention at 6-month follow-up (+0.48 serves vs. +0.43 serves [27], respectively). The replication of this effect was achieved despite intervention adaptations, which included additional content to promote healthy sleep and movement behaviors. The effect size of this outcome for the randomized participants is surprising given that the benefits of health interventions are usually attenuated as they move from trials undertaken under optimal conditions to more ‘real-world’ conditions, such as the case in this study [40,41,42]. Further, the significant difference in parent vegetable consumption found in the current study (up to 0.48 serves) is particularly encouraging as less than 1 in 10 Australian adults meet the recommended intakes for vegetables per day (at least 5 serves/day) [43], and identification of strategies that can increase adult vegetable consumption remains a key public health priority. Conversely, this trial did not see changes in parent fruit consumption for any of the analyses, an effect that may be due to a ‘ceiling effect’ [44], in that average parent intakes of fruit were already close to meeting the recommendations (i.e., 2 serves/day) at baseline (see Table 4). 

It is noteworthy that the only statistically significant results were found for parents randomized to receive the telephone intervention. This was the modality selected least frequently by parents when allocated based on preference. It should be noted, however, that differences in participant demographics between the randomized and preference participants may have influenced this finding [32]. Additionally, baseline vegetable intakes were higher in parents randomly allocated (~3.1 serves/day) compared with parents allocated based on preference (~2.7 serves/day). However, our analyses adjusted for baseline values. 

More parents expressed a preference for the online intervention, but there was a lower level of engagement for this intervention modality (online completion rate: 26% vs. telephone completion rate: 33%) [32]. Given participant engagement remains a barrier for technology-based health interventions [45], it is important that future research focuses on better understanding the barriers and enablers to parent engagement with technology-based interventions to optimize improvements to family lifestyle behaviors. Additionally, further comparative effectiveness studies are needed that compare telephone-based interventions to other broad-reaching intervention modalities (e.g., tailored printed and online programs) to determine if efficacious interventions can be implemented in ‘real-world’ contexts, and achieve similar outcomes to those observed in controlled settings [46].

The current study demonstrates that parent diet can improve following participation in telephone interventions targeting the dietary behaviors of their children. This is an important contribution to the literature, given that mothers’ vegetable consumption behaviors and preferences are thought to be a major determinant of child vegetable intake [25]. Targeting parents of children may facilitate better dietary outcomes for parents, as parent beliefs that their behaviors (i.e., modeling healthy eating) can influence the health of their child are a strong motivator for behavior change [47]. ‘Time for Healthy Habits’ actively engaged only one parent (predominantly the mother) of the household, but both parents have been shown to have a significant influence on young children’s food intake [48], and future interventions may benefit from encouraging active intervention engagement from all parents within the household. The increases in parent vegetable consumption found as a result of parent participation in the telephone intervention were not mirrored in child vegetable consumption (reported elsewhere) [32]. However, the telephone intervention was found to reduce child non-core food intake [32]. This suggests that changes to child vegetable intakes may require more time, particularly given evidence that new foods need to be offered multiple times before they are accepted [49]. 

The findings from this study should be considered in light of the limitations and strengths. This novel study adds to the limited published translational research by adapting and scaling up two efficacious interventions to enable delivery across NSW, Australia. Although parent fruit and vegetable intakes were based on self-reported data, survey items have been previously validated against fruit and vegetable biomarkers [38] and have been frequently used to collect population data, which may enable comparison with state-wide datasets [36]. All interventions were delivered remotely, which enabled wider reach, and greater access and equity for participants located outside of metropolitan locations. 

Limitations of this trial include the low number of participants with no preference for their intervention delivery modality, and the need to apply a stopping rule to ensure adequate power among randomized participants in the main analysis. Prior to this stopping rule being enacted, there were large imbalances in the distribution of participant preferences across groups (preference for telephone n = 22; online n = 132; written n = 64), and so the effect size among parents allocated based on preference should be interpreted with caution, as low numbers in the telephone group may have limited our ability to conduct a robust analysis. This study sought to compare the effectiveness of two existing intervention packages. While they both aimed to increase child fruit and vegetable consumption, and utilized some similar content and components, they were discrete interventions, and this study was not intended to identify whether the online or telephone delivery channel was more effective, but rather which of the two packages was more effective. This study was driven by a pragmatic research question that sought to identify which of the two existing intervention packages would be more effective if delivered as a state-wide service to the people of NSW. It was outside the scope of the current study to explore the effectiveness of each intervention based on allocation (i.e., between participants who had been allocated randomly or allocated based on preference to a group). As such, we did not compare the effectiveness of participants randomized to receive the telephone-based intervention, with participants allocated to the telephone-based intervention based on preference. Similarly, it was outside the scope of the study to directly compare intervention delivery mechanisms (telephone vs. online). Future research should investigate how participant preferences interact with intervention effectiveness in a larger sample. This interaction was beyond the current scope of this translational trial, which posed separate questions about participant preferences for service type, and about the effectiveness of three different interventions [32].

Although the parallel partially randomized patient preference trial design was selected to better replicate the implementation of dissemination strategies within real-world contexts, and offer better external validity than traditional RCTs [31], university-educated parents were over-represented in our sample. The ‘Time for Healthy Habits’ trial was conducted between May 2019 and March 2020. This period coincided with the NSW 2019–2020 summer bushfires, as well as the first COVID-19 outbreak in Australia, and it is highly likely that intervention effects and parent engagement were interrupted due to this. Finally, although attrition rates for the 9-month follow-up were relatively high (i.e., randomly allocated: 23–34%; preference allocated: 18–42%), they are consistent with other remotely delivered interventions [45]. To explore potential attrition bias, the analysis included both complete-case and multiple imputation approaches, and results were found to be consistent between the two approaches.

## 5. Conclusions

This study reports the effectiveness of two remotely delivered healthy lifestyle interventions (telephone and online) that targeted the parents of 2- to 6-year-old children in increasing parent fruit and vegetable intakes (secondary outcomes for the ‘Time for Healthy Habits’ RCT). Study results indicate that parents who were randomized to receive the telephone intervention significantly increased their vegetable but not fruit consumption, relative to controls receiving written materials. There may be some benefit to parents from participating in the Healthy Habits Plus (telephone-based) intervention aimed at improving the eating behaviors of their children. However, given the relatively small sample size, ongoing evaluation within a larger sample of participants is required and future research is recommended to explore methods for optimizing parent engagement with technology-based interventions, to enable greater health benefits for both parents and their children. This analysis of secondary outcomes adds to the primary trial outcomes [32], emphasizing that participant preferences, intervention engagement and intervention effectiveness do not necessarily align, representing an ongoing challenge for policymakers when deciding which population health services should be prioritized for investment.

## Figures and Tables

**Figure 1 ijerph-19-06165-f001:**
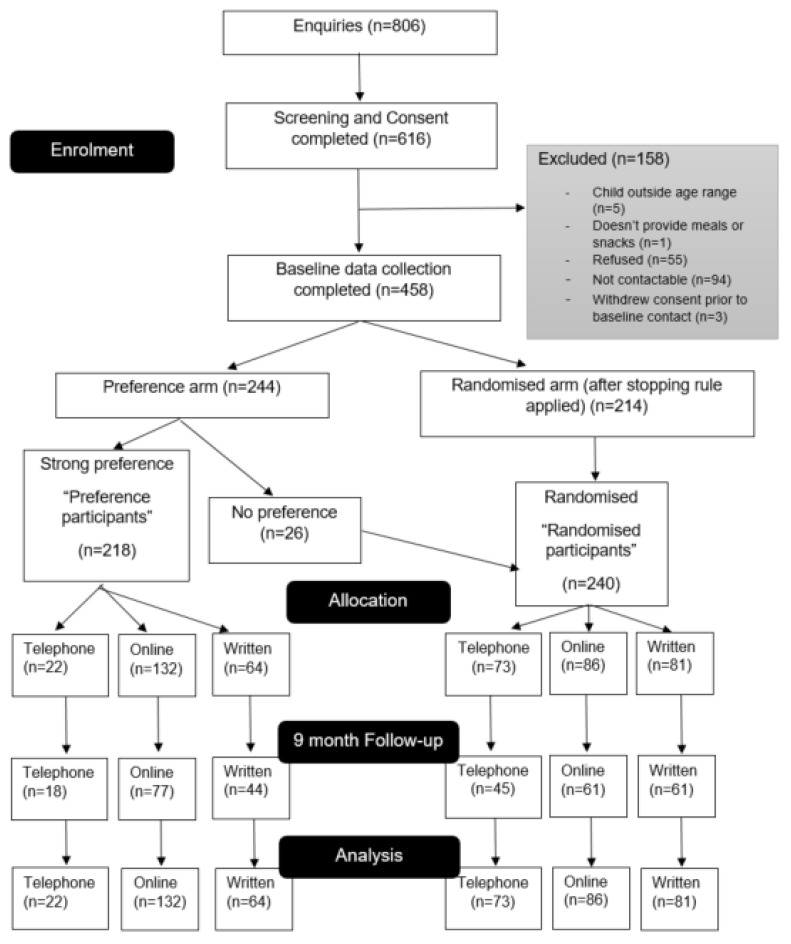
CONSORT flow diagram [32].

**Table 1 ijerph-19-06165-t001:** Summary of interventions delivered as part of the ‘Time for Healthy Habits’ trial.

Intervention	Telephone Intervention (Healthy Habits Plus)	Online Intervention (Time2bHealthy)
Delivery Mode	Telephone-based Delivered by trained para-professionals	Online web application (mobile optimized) Online components moderated by a health professional
Intervention Components	Six motivational support calls (approx. 20–30 min each) Delivered fortnightly over approximately 3 months Printed guidebook and pad of menu planners mailed to parents, and used during the telephone contracts	Six online modules (approx. 30 min each) Delivered over approximately 3 months (i.e., one module every 2 weeks) Closed Facebook group (optional)
Intervention Content	Both programs sought to improve healthy eating and movement behaviors (physical activity, sedentary screen time and sleep), and focused on: The availability and accessibility of foods and beverages (i.e., ensuring fruit and vegetables are present and stored in a ready-to-eat form) and opportunities for physical activity in the home and limiting the presence of screens/devices; Supportive family routine (i.e., eating meals without the television, having a set bedtime); Parental role modeling of health behaviors (i.e., demonstrating enjoyment of eating healthy foods).	
Adherence Strategies	Up to 10 attempt calls were made to participants at each scheduled call. Unsuccessful call attempts were complemented by SMS and email reminders, to encourage participants to schedule a convenient time to receive their next telephone call to complete the intervention module.	Participants received an email each week reminding them to long onto the website to complete the modules. 1–2 moderator posts were made on the Facebook group each week to remind participants to long onto the website and contribute to Facebook discussion.

**Table 2 ijerph-19-06165-t002:** Parent characteristics at baseline [32].

	Randomized Participants (n = 240)			Preference Participants (n = 218)			ALL
	Telephone	Online	Written Control	Telephone	Online	Written Control	
	n = 73	n = 86	n = 81	n = 22	n = 132	n = 64	n = 458
Age, in years							
Mean	34.9	36.6	36.8	37.2	36.1	35.8	36.1
SD	4.5	4.9	5.1	5.5	4.8	5.0	4.9
Sex—female							
N	69	83	79	22	126	62	441
%	94.5	96.5	97.5	100	95.5	96.9	96.3
University-educated							
N	57	64	53	20	88	40	322
%	78.1	74.4	65.4	90.9	66.7	62.5	70.3
Annual household income > AUD 100,000							
N	49	58	57	16	91	40	311
%	67.1	69.0	71.3	72.7	68.9	62.5	68.3

SD: standard deviation; N: frequency; %: proportion.

**Table 3 ijerph-19-06165-t003:** Changes in parent consumption of vegetables (mean daily serves).

	Baseline Intake	Follow-Up Intake	Complete Case Analysis ^a^	Multiple Imputation Analysis ^b^
	Mean (SD)	Mean (SD)	Mean Difference vs. Control (95% CI), *p*-Value	Mean Difference vs. Control (95% CI), *p*-Value
**Randomized participants**				
Telephone intervention (n = 73)	3.12 (1.89)	3.58 (1.37)	0.48 (0.07, 0.88), ***p*** **= 0.02**	0.41 (0.02, 0.81), ***p*** **= 0.04**
Online intervention (n = 86)	2.88 (1.44)	3.20 (1.53)	0.24 (−0.13, 0.61), *p* = 0.20	0.24 (−0.13, 0.61), *p* = 0.21
Written control (n = 81)	3.23 (1.65)	3.02 (1.20)	Reference	Reference
**Preference participants**				
Telephone intervention (n = 22)	2.36 (1.14)	2.50 (1.15)	−0.19 (−0.77, 0.38), *p* = 0.51	−0.14 (−0.70, 0.42), *p* = 0.62
Online intervention (n = 132)	3.05 (1.41)	3.23 (1.28)	0.05 (−0.34, 0.44), *p* = 0.80	0.11 (−0.26, 0.48), *p* = 0.56
Written control (n = 64)	2.78 (1.31)	2.91 (1.44)	Reference	Reference
**All participants**				
Telephone intervention (n = 95)	2.94 (1.77)	3.25 (1.39)	0.27 (−0.06, 0.60), *p* = 0.11	0.27 (−0.06, 0.60), *p* = 0.1
Online intervention (n = 218)	2.98 (1.42)	3.22 (1.42)	0.15 (−0.12, 0.42), *p* = 0.26	0.17 (−0.09, 0.44), *p* = 0.2
Written control (n = 145)	3.03 (1.52)	2.98 (1.30)	Reference	Reference

^a^ Complete case analysis: included data for parents that reported baseline and follow-up data (n = 306); ^b^ Multiple imputations analysis: follow-up data for imputed for parents with missing follow-up data (n = 458). Boldface indicates statistical significance (*p* < 0.05).

**Table 4 ijerph-19-06165-t004:** Change in parent consumption of fruit (mean daily serves).

	Baseline Intake	Follow-Up Intake	Complete Case Analysis ^a^	Multiple Imputation Analysis ^b^
	Mean (SD)	Mean (SD)	Mean Difference vs. Control (95% CI), *p*-Value	Mean Difference vs. Control (95% CI), *p*-Value
**Randomized participants**				
Telephone intervention (n = 73)	1.71 (0.98)	1.89 (0.88)	0.08 (−0.25, 0.42), *p* = 0.62	−0.05 (−0.38, 0.27), *p* = 0.75
Online intervention (n = 86)	1.85 (0.98)	1.70 (1.12)	−0.16 (−0.46, 0.15), *p* = 0.32	−0.02 (−0.37, 0.34), *p* = 0.92
Written control (n = 81)	1.68 (1.00)	1.73 (0.92)	Reference	Reference
**Preference participants**				
Telephone intervention (n = 22)	1.76 (1.09)	1.78 (1.00)	0.06 (−0.42, 0.54), *p* = 0.79	−0.05 (−0.64, 0.55), *p* = 0.88
Online intervention (n = 132)	1.86 (0.97)	1.91 (0.99)	0.15 (−0.18, 0.48), *p* = 0.37	−0.07 (−0.49, 0.34), *p* = 0.72
Written control (n = 64)	1.68 (0.82)	1.64 (0.97)	Reference	Reference
**All participants**				
Telephone intervention (n = 95)	1.72 (1.00)	1.86 (0.91)	0.09 (−0.18, 0.36), *p* = 0.51	−0.05 (−0.31, 0.21), *p* = 0.68
Online intervention (n = 218)	1.85 (0.97)	1.82 (1.05)	0.00 (−0.22, 0.22), *p* = 1.00	−0.05 (−0.28, 0.18), *p* = 0.67
Written control (n = 145)	1.68 (0.92)	1.69 (0.94)	Reference	Reference

^a^ Complete case analysis: included data for parents that reported baseline and follow-up data (n = 306); ^b^ Multiple imputations analysis: follow-up data for imputed for parents with missing follow-up data (n = 458).

## Data Availability

The data are not publicly available as data sharing was not sought in the ethics application. Therefore, ethics approval has been obtained for data to be used for this study only.

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
