# Peer review of "Parent Fruit and Vegetable Consumption Outcomes from the Translational ‘Time for Healthy Habits’ Trial: Secondary Outcomes from a Partially Randomized Preference Trial"

_ijerph, 2022, doi:10.3390/ijerph19106165_

Round 1

Reviewer 1 Report

This study demonstrates the effectiveness of two remote healthy lifestyle interventions (telephone and online) targeting parents of children 2 to 6 years of age in increasing their parents' fruit and vegetable consumption. Research results show that only parents who were randomized to telephone intervention significantly increased their consumption of vegetables.

Comments and suggetions:

When using the abbreviation for the first time in the text of the work, its full and abbreviated name should be given (NSW – line 35, RCT – line 131, CATI – line 217).

L40. '… (95% CI: 0.07, 0.88)…' too detailed data in the abstract. I propose to remove.

L53. 400 gm? SI units should be used.

L59-70. Repetitions, needs improvement. Rephrase.

L149-158. Incorrect line spacing.

L317-321. How should the differences in vegetable consumption between the participants (randomized or preference participants) be interpreted? What could be the reason for this? And how does this affect future research? Please explain.

L320. 75gm? SI units should be used.

Conclusions: Please add a comment about preference participants for either the telephone or online intervention in the consumption of  fruit and vegetables.

Author Contributions: use the authors' initials instead of the full name.

References: incorrect style and format (see Instructions for Authors of IJERPH); needs to be improved.

Author Response

Comment 1.1: This study demonstrates the effectiveness of two remote healthy lifestyle interventions (telephone and online) targeting parents of children 2 to 6 years of age in increasing their parents' fruit and vegetable consumption. Research results show that only parents who were randomized to telephone intervention significantly increased their consumption of vegetables.

When using the abbreviation for the first time in the text of the work, its full and abbreviated name should be given (NSW – line 35, RCT – line 131, CATI – line 217).

Response 1.1: Thank you for identifying this. Please find abbreviations for NSW and RCT have been added at first mention in text. The abbreviation for CATI was already provided in section 2.3, page 4.

Comment 1.2: L40. '… (95% CI: 0.07, 0.88)…' too detailed data in the abstract. I propose to remove.

Response 1.2: Please find the reported 95% confidence interval has been removed from the abstract.

Comment 1.3: L53. 400 gm? SI units should be used.

Response 1.3: Please find grams are now reported in SI units (i.e. g)

Comment 1.4: L59-70. Repetitions, needs improvement. Rephrase.

Response 1.4: Please find this section has been rephrased to reduce repetition.

“Early childhood is an important period to prioritize good nutrition [1,2] as the eating habits adopted by children early in life track throughout the lifespan and into adulthood [3-5]. Parents and carers (hereafter ‘parents’) of young children play a critical role in creating a supportive home food environment through the foods that they make available and accessible to their children[6] and their role modeling of healthy eating behaviors [7]. Parent-targeted interventions which focus on changing the home food environment of young children are effective in establishing healthy child dietary habits [17,18], and could be an important public health strategy to minimize the future burden of diet-related chronic diseases.”

Comment 1.5: L149-158. Incorrect line spacing.

Response 1.5: Please find line spacing has been corrected.

Comment 1.6: L317-321. How should the differences in vegetable consumption between the participants (randomized or preference participants) be interpreted? What could be the reason for this? And how does this affect future research? Please explain.

Response 1.6: Although the number of total randomised vs preference parents overall were relatively similar, the distribution between intervention groups were quite different as the randomised parents were allocated in an approximate equal ratio (Telephone n=73; Online n=86; Control n=81), however preference parents heavily favoured the online group (Telephone n=22; Online n=132; Control n=64), highlighting that only a small number of parents were in the telephone group based on preference. The implications of this have been explained in the limitations section of the discussion:

 “Limitations of this trial include the low number of participants with no-preference for their intervention delivery modality, and the need to apply a stopping rule to ensure adequate power among randomized participants in the main analysis. Prior to this stopping rule being enacted, there were large imbalances in the distribution of participant preferences across groups (preference for telephone n=22; online n=132; written n=64), and so the effect size among parents allocated based on preference should be interpreted with caution, as low numbers in the telephone group may have limited our ability to conduct a robust analysis.”

“Future research should investigate how participant preferences interact with intervention effectiveness in a larger sample. This interaction was beyond the current scope of this translational trial, which posed separate questions about participant preferences for service type, and about the effectiveness of three different interventions [32]”.

Comment 1.7: L320. 75gm? SI units should be used.

Response 1.7: Please find grams are now reported in SI units (i.e. g)

Comment 1.8: Conclusions: Please add a comment about preference participants for either the telephone or online intervention in the consumption of fruit and vegetables.

Response 1.8: The authors were cautious to draw too strong of conclusions for the preference participants, particularly given the small sample size of telephone preference participants (n=22).

As mentioned in our limitations:

Prior to this stopping rule being enacted, there were large imbalances in the distribution of participant preferences across groups (preference for telephone n=22; online n=132; written n=64), and so the effect size among parents allocated based on preference should be interpreted with caution, as low numbers in the telephone group may have limited our ability to conduct a robust analysis.”

However, please find we now elaborated on these ideas within our conclusion:

“There may be some benefit to parents from participating in the Healthy Habits Plus (telephone-based) intervention aimed at improving the eating behaviors of their children. However, given the relatively small sample size, ongoing evaluation within a larger sample of participants is required and future research is recommended to explore methods for optimizing parent engagement with technology-based interventions, to enable greater health benefits for both parents and their children. This analysis of secondary outcomes adds to the primary trial outcomes [32], emphasizing that participant preferences, intervention engagement and intervention effectiveness do not necessarily align, representing an ongoing challenge for policy-makers when deciding which population health services should be prioritized for investment.”

Comment 1.9: Author Contributions: use the authors' initials instead of the full name.

Response 1.9: Please find author initials have now been used.

Comment 1.10: References: incorrect style and format (see Instructions for Authors of IJERPH); needs to be improved.

Response 1.10: Thank you for highlighting this error. Please find we have updated the reference style and format as per IJERPH specifications.

Reviewer 2 Report

The manuscript had several flaws that need to be improved for better clarity to the audience

1) The abstract must be rewritten more clearly 

2) Line 129, "Participants in the randomized arm were randomly allocated in a 1:1:1 ratio by an independent statistician" but still in each intervention the N is not the same.  Kindly rewrite and elaborate on it.

3) Line 269 to 272 to be rewritten.

4) Telephone intervention group received  Printed, hardcopy resources (i.e., a guidebook and menu planner) why it is used in telephone group and not in the online group and it may have an impact on the outcomes? 

5) How printed hard copy resources in telephone intervention group differ from the control group written educational materials developed by the NSW Office of Preventive Health?

6) I strongly disagree with the authors regarding their result and conclusion states that children’s healthy lifestyle interventions delivered to parents via the telephone can have broader public health nutrition impacts, highlighting the potential co-benefits for parent and child dietary intakes where their N is too small especially in telephone group to come into a conclusion. I need still more participants are required to come up with a strong conclusion. 

7) Did the authors take any measures to nullify the celling effect?

8) Line 339 reference required "Fruit 
were already close to meeting the recommendations (i.e., 2 serves/day) at baseline"

(9) The statistically significant results were found for parents 
randomized to receive the telephone intervention because the telephone intervention group was called and reminded more frequently than (Up to 10 attempt calls were made to participants at each scheduled call and unsuccessful call attempts were complemented by SMS and email reminders) compared to the online intervention group (one reminder each week). In addition, the randomized telephone group has a comparatively lower female % ratio (94%) compared to the other two groups. The authors may engage all participants as the female sex to avoid variability caused by male subjects

Author Response

Comment 2.1: The manuscript had several flaws that need to be improved for better clarity to the audience

  • The abstract must be rewritten more clearly 

Response 2.1. Please find we have attempted to improve the clarity of the abstract:

“Healthy eating and active living interventions targeting parents of young children could have benefits for both children and parents. The aim of this study was to assess the effectiveness of two remotely delivered healthy eating and active living interventions delivered at scale to parents, in increasing parent vegetable and fruit consumption (pre-specified secondary outcomes). Parents of children aged 2-6 years residing in New South Wales Australia (n=458) were recruited to a partially randomized preference trial consisting of three groups (Telephone intervention (n=95); Online intervention (n=218); Written material (Control) (n=145)). This design allowed parents with a strong preference to select their preferred intervention and once preference trends had been established, all parents that were subsequently recruited were randomized to obtain robust relative effects. Parent vegetable and fruit consumption was assessed via telephone interview at baseline and 9-months later. At follow-up, randomized parents who received the telephone intervention (n=73) had significantly higher vegetable consumption compared to those who received the written control (n=81) (+0.41 serves/day, p=0.04), but there were no differences in parents allocated to intervention group based on preference. No differences in fruit consumption were found for randomized or preference participants for either the telephone or online intervention. There may be some benefit to parents participating in the Healthy Habits (telephone-based) intervention aimed at improving the eating behaviors of their children.”

Comment 2.2: Line 129, "Participants in the randomized arm were randomly allocated in a 1:1:1 ratio by an independent statistician" but still in each intervention the N is not the same.  Kindly rewrite and elaborate on it.

Response 2.2:  Please find we have rephrased the process for randomisation to improve accuracy.

“Participants in the randomized arm were randomly allocated using a random number function generated by an independent statistician, whereby each of the three arms were given a 33% likelihood of being selected.”

Comment 2.3: Line 269 to 272 to be rewritten.

Response 2.3: The authors are unable to identify the text this reviewer is referring to, as the provided manuscript does not contain line numbers. However, the authors will be happy to try to accommodate this request upon further clarification.

Comment 2.4: Telephone intervention group received printed, hardcopy resources (i.e., a guidebook and menu planner) why it is used in telephone group and not in the online group and it may have an impact on the outcomes? 

Response 2.4: To clarify, we were not testing the exact same intervention strategies or content delivered via different channels. This trial tested two different intervention packages (each with previous demonstrated efficacy in separate prior RCTs), that both targeted parents of children 2-6 year-olds, to improve child diet outcomes. One was delivered primarily via the telephone and one was delivered online.

As we highlight in text and in table 1, there are some key similarities between the two interventions, but there are also some key differences between the interventions other than their mechanism of delivery. We have simply labelled the interventions as the ‘telephone intervention’ and the ‘online intervention’ for reader ease and consistency between published manuscripts.

Comment 2.5: How printed hard copy resources in telephone intervention group differ from the control group written educational materials developed by the NSW Office of Preventive Health?

Response 2.5: As specified in text, the telephone intervention group received a printed guidebook, and menu planner (developed by the research team), which allowed the participants to follow the intervention content with the interventionists over the phone. This written material was intended to be used in conjunction with the calls.

The control group received written fact sheets that had been developed by the NSW Office of Preventive Health, and did not contain any activities. As mentioned in text, the fact sheet topics included: Healthy breakfasts, Healthy lunchboxes, Positive family mealtime, Healthy drinks, Fussy eating, Encouraging children to eat vegetables and fruit; Healthy snacks; Active play; Reducing screen time, and; Encouraging healthy sleep habits. The fact sheets were delivered as standalone resources.

Comment 2.6: I strongly disagree with the authors regarding their result and conclusion states that children’s healthy lifestyle interventions delivered to parents via the telephone can have broader public health nutrition impacts, highlighting the potential co-benefits for parent and child dietary intakes where their N is too small especially in telephone group to come into a conclusion. I need still more participants are required to come up with a strong conclusion. 

Response 2.6: Please find we have updated our conclusion to address this concern.

“There may be some benefit to parents from participating in the Healthy Habits Plus (telephone-based) intervention aimed at improving the eating behaviors of their children. However, given the relatively small sample size, ongoing evaluation within a larger sample of participants is required and future research is recommended to explore methods for optimizing parent engagement with technology-based interventions, to enable greater health benefits for both parents and their children.”

Comment 2.7:  Did the authors take any measures to nullify the celling effect?

Response 2.7: No measures were taken to nullify the ceiling effect. This was a public health intervention for the general population. As such, we did not attempt to identify and intervene specifically with those with low fruit consumption.

Comment 2.8:  Line 339 reference required "Fruit 
were already close to meeting the recommendations (i.e., 2 serves/day) at baseline"

Response 2.8: Please find a reference has been added.

Conversely, this trial did not see changes in parent fruit consumption for any of the analyses, an effect that may be due to a ‘ceiling effect’ [8], in that average parent intakes of fruit were already close to meeting the recommendations (i.e., 2 serves/day) at baseline (see table 4).”

Comment 2.9: The statistically significant results were found for parents 
randomized to receive the telephone intervention because the telephone intervention group was called and reminded more frequently than (Up to 10 attempt calls were made to participants at each scheduled call and unsuccessful call attempts were complemented by SMS and email reminders) compared to the online intervention group (one reminder each week). In addition, the randomized telephone group has a comparatively lower female % ratio (94%) compared to the other two groups. The authors may engage all participants as the female sex to avoid variability caused by male subjects

Response 2.9: The reviewer makes some interesting points.

To be clear, this trial tested two different intervention packages (each with previous demonstrated efficacy in separate prior RCTs), that both targeted parents of children 2-6 year-olds, to improve child diet outcomes. One was delivered primarily via the telephone and one was delivered online. As we highlight in text and in table 1, there are some key similarities between the two interventions, but there are also some key differences between the interventions other than their mechanism of delivery. We have simply labelled the interventions as the ‘telephone intervention’ and the ‘online intervention’ for reader ease and consistency between published manuscripts.

To highlight this point, please find we have updated our study conclusions to reflect the fact that our intervention ‘packages’ (i.e. Healthy Habits Plus; Time2bHealthy) are not the same as the intervention delivery mechanism (i.e. telephone-based; online).

“There may be some benefit to parents from participating in the Healthy Habits Plus (telephone-based) intervention aimed at improving the eating behaviors of their children.”

With regards to differences in participant characteristics between groups, it is important to note that this was a translation trial attempting to provide an indication of the effectiveness of the interventions in a real-world scenario. As such, we deliberately made no attempt to ensure we have a homogenous sample- as we wanted our sample to reflect real world users of such services. Although our intervention participants were primarily mothers, this finding is consistent with other public health parent support services (e.g. Parentline interviewed primarily mothers called about one or more of their children Microsoft Word - Parentline Plus.doc (ucl.ac.uk)).

Reviewer 3 Report

The reviewed manuscript is an interesting and well-written paper assessing the effectiveness of the telephone and online interventions in changing parent fruit and vegetable consumption during a follow-up visit during the “Time for Healthy Habits” Trial. The authors provide a well-organized structure, an accurate scientific tone, an efficient analysis of the results and a relevant Discussion section.

My only comments are related to a few formal matters. The authors should include in the Abstract an explanation about this analysis being a pre-specified secondary outcome in the trial mentioned above and should re-check and uniformise the position of first-name initials across the References section.

Author Response

Comment 3.1: The reviewed manuscript is an interesting and well-written paper assessing the effectiveness of the telephone and online interventions in changing parent fruit and vegetable consumption during a follow-up visit during the “Time for Healthy Habits” Trial. The authors provide a well-organized structure, an accurate scientific tone, an efficient analysis of the results and a relevant Discussion section.

Response 3.1: The authors are most grateful for this reviewers considered comments and feedback, and are pleased to hear that this reviewer found this manuscript to be well-organized, with an efficient analysis, accurate scientific tone and relevant discussion section.

Comment 3.2: My only comments are related to a few formal matters. The authors should include in the Abstract an explanation about this analysis being a pre-specified secondary outcome in the trial mentioned above and should re-check and uniformise the position of first-name initials across the References section.

Response 3.2: Thank you for making these suggestions. Please find our abstract now includes:

The aim of this study was to assess the effectiveness of two remotely delivered healthy eating and active living interventions delivered at scale to parents, in increasing parent fruit and vegetable consumption (pre-specified secondary outcomes).”

Please find we have also updated our referencing style to reflect journal specifications.

Round 2

Reviewer 2 Report

.

Author Response

Reviewer 2 comment: The number of participants (n) is too small to come to a conclusion.

Response to reviewer 2:

The study conclusions relate to the main analysis of randomised participants, of which there were 240 in total, with a minimum group size of 73 (telephone group). Despite this relatively modest sample size we found a statistically significant difference favouring parents randomised to the telephone-based intervention.

We note in the latest issues listed on the IJERPH website multiple RCTs with much lower sample sizes than the current study, which have drawn conclusions based on the available data.

  • Effectiveness of App-Based Intervention to Improve Health Status of Sedentary Middle-Aged Males and Females (RCT with 35 subjects in total)
  • Efficacy of Ultrasound-Guided Percutaneous Lavage and Biocompatible Electrical Neurostimulation, in Calcific Rotator Cuff Tendinopathy and Shoulder Pain, A Prospective Pilot Study (RCT with 40 patients)
  • (Home-Based Resistance Training for Older Subjects during the COVID-19 Outbreak in Italy: Preliminary Results of a Six-Months RCT (RCT with 9 participants)

We also highlight in the manuscript that the Time for Healthy Habits trial was a real-world translational trial that was running through the original COVID-19 outbreak in Australia, and under these circumstances 73 participants actually represents a reasonable participant number.

However, we would like to highlight that we have previously listed sample size as a potential study limitation at multiple points throughout the manuscript; and we have refrained from drawing conclusions about the preference-based participants, where despite a total sample of 218 participants, the numbers in the telephone arm were quite low (n=22). “The effect size among parents allocated based on preference should be interpreted with caution, as low numbers in the telephone group (n=22) may have limited our ability to conduct a robust analysis”

Furthermore, the analysis of the preference-based participants (i.e. non-randomised) is not the main analysis of the paper. However, as this analysis was pre-specified, as per best-practice, we have also reported it alongside the main analysis of randomised participants. We do not attempt to make broad conclusions based on the preference-based results without pointing out the limitations. 

Additionally, please find our conclusion has already been modified to incorporate the reviewer concerns and reads as follows: “There may be some benefit to parents from participating in the Healthy Habits Plus (telephone-based) intervention aimed at improving the eating behaviours of their children. However, given the relatively small sample size, ongoing evaluation within a larger sample of participants is required and future research is recommended to explore methods for optimizing parent engagement with technology-based interventions, to enable greater health benefits for both parents and their children”.